# The Role of Light-Regulated Auxin Signaling in Root Development

**DOI:** 10.3390/ijms24065253

**Published:** 2023-03-09

**Authors:** Fahong Yun, Huwei Liu, Yuzheng Deng, Xuemei Hou, Weibiao Liao

**Affiliations:** College of Horticulture, Gansu Agricultural University, 1 Yingmen Village, Anning District, Lanzhou 730070, China; yunfh@st.gsau.edu.cn (F.Y.); lhwgsauedu@163.com (H.L.); hxm1511704@163.com (X.H.)

**Keywords:** light-responsive signaling pathways, root growth and development, target genes and proteins, root negative phototropism, gravitropism

## Abstract

The root is an important organ for obtaining nutrients and absorbing water and carbohydrates, and it depends on various endogenous and external environmental stimulations such as light, temperature, water, plant hormones, and metabolic constituents. Auxin, as an essential plant hormone, can mediate rooting under different light treatments. Therefore, this review focuses on summarizing the functions and mechanisms of light-regulated auxin signaling in root development. Some light-response components such as phytochromes (PHYs), cryptochromes (CRYs), phototropins (PHOTs), phytochrome-interacting factors (PIFs) and constitutive photo-morphorgenic 1 (COP1) regulate root development. Moreover, light mediates the primary root, lateral root, adventitious root, root hair, rhizoid, and seminal and crown root development via the auxin signaling transduction pathway. Additionally, the effect of light through the auxin signal on root negative phototropism, gravitropism, root greening and the root branching of plants is also illustrated. The review also summarizes diverse light target genes in response to auxin signaling during rooting. We conclude that the mechanism of light-mediated root development via auxin signaling is complex, and it mainly concerns in the differences in plant species, such as barley (*Hordeum vulgare* L.) and wheat (*Triticum aestivum* L.), changes of transcript levels and endogenous IAA content. Hence, the effect of light-involved auxin signaling on root growth and development is definitely a hot issue to explore in the horticultural studies now and in the future.

## 1. Introduction

Plant growth and development includes seed germination, nutrient formation, reproductive formation, flowering, pollination and fertilization, and seed setting [1]. The factors that influence plant growth and development are various, including light, temperature, water, air, soil, nutrition, hormones and pathogens [2]. Among these factors, the role of the light environment is one of the most essential factors. Light is involved in all stages of plant growth and development, including seed dormancy and germination, seedling establishment, seedling etiolation, cotyledon opening, hypocotyl elongation, root development and flowering [3,4]. For example, photoperiodic lighting promoted plant growth and shortened flowering time only in early flowering in *Ranunculus asiaticus* L., and the effect was stronger under red:far red (R:FR) 3:1 light [5]. The role of light in seed germination has also been widely reported in the past 20 years. Light promotes the seed germination rate in wild grain (*Brachypodium disachyon)*, and the positive role is offset by subsequent FR light exposure, suggesting that photochromes might be involved in seed germination in *B. distachyon* [6]. Moreover, different species have different light perception during seed germination of barley (*Hordeum vulgare* L.) and wheat (*Triticum aestivum* L.) have different light perception during seed germination [7]. White and red light was able to enhance seed germination in *Arabidopsis thaliana*, while white and blue light promoted dormancy in barley and wheat. Hence, the mechanism of light-mediated plant growth and development is complex.

The root is a key storage organ for water and carbohydrates that can consolidate soil and absorb water, oxygen, nutrients and minerals [8]. However, plant rooting is affected by light quality, quantity and direction [9]. As we know, light has a significant effect on root physiology and development. Van Gelderen et al. [10] found that supplementing white light with FR light reduced lateral root density. Moreover, roots were able to coordinate their responses to light through the expression of hypocotyl 5 (HY5) [10]. HY5 accumulated in lateral root primordium (LRP) and the cortex under FR light irradiation [10].

Similarly, plant hormones such as ethylene (ETH), cytokinins (CTKs) and abscisic acid (ABA) act as important extrinsic factors in root development [11]. Auxin is one of the most important phytohormones, and plays a major role in root development. Light is also involved in the auxin signaling pathway in root development [10,12,13]. For example, ultraviolet-B (UV-B) photoreceptor UVR8 directly interacted with MYB transcriptional factors MYB domain protein 73/77 (MYB73/MYB77) to regulate auxin responses and lateral root growth [12]. Furthermore, van Gelderen et al. [13] observed that the protein expression of auxin transporters PIN3 and LAX3 was reduced in surfacing cortical cells, indicating the crosstalk between light and auxin signaling during plant rooting. Transmembrane kinase 1 (Tmk1) played an essential role in the differential regulation of auxin recruitment to the clathrin of light chains (CLCs) and clathrin heavy chains (CHCs) in *A. thaliana*, which influenced the polar distribution of PIN2 and the asymmetric distribution of auxin, thus finally affecting root gravitropic growth [14].

Light regulates root development directly or indirectly by modifying auxin biosynthesis and/or signaling. The crosstalk between light and auxin signaling for shoot–root communication has been summarized by Halliday et al. [15]. Recently, Yang and Liu [16] summarized the progress relation to understanding how shoots and roots coordinate their responses to light through different light-signaling components and pathways. However, the molecular mechanism of light–auxin crosstalk during rooting, phototropism, and geotropism has not been systematically reviewed. Additionally, the target genes or proteins of the light response to auxin-signaling transduction for rooting have not been summarized to date. Therefore, in this review, we provide an overview of how the crosstalk between light and auxin signaling regulates plant root development. We list the target genes or proteins and explore the molecular mechanism of light-mediated root growth and development through the auxin-signaling pathway, to provide theoretical references for the application of light and auxin in root growth and development in horticultural products.

## 2. Light Signaling in Root Development

### 2.1. Light Perception

Plants have a wide spectrum of light-absorption response, ranging from near-UVB (280–315 nm) to FR (~750 nm) wavelengths. The types of UV radiation perception are various: cryptochromes (CRYs), phototropins (PHOTs), zeitlupes (ZTLs), flavin-binding Kelch repeat F-box (FKF) and Lov Kelch repeat Protein 2 (LKP2) perceive blue/UV-A light (315–400 nm) [1]; and UV resistance locus 8 (UVR8) perceives UV-A and UV-B (280–315 nm) light [17,18]. The apolipoprotein of photopigment is synthesized in cytoplasm, then it is attached to a linear tetrapyrin chromophore, called phytochrome, to produce the red-absorbing pigment Pr. Upon exposure to red light, the Pr form is converted to an active, FR light-absorbing form (Pfr) and transferred to the nucleus [19]. After sensing various light wavelengths, these photoreceptors further transmit signals through a cascade to regulate the expression of different genes, resulting in physiological responses [20].

### 2.2. Photoreceptors Involved in Root Development

Various photoreceptors have been reported, including phytochromes (PHYs), cryptochromes (CRYs), phototropins (PHOTs) and UVR8. Even roots, which are normally not exposed to light, express photoreceptors and can respond to light by developing chloroplasts. PHYs are dimeric chromoproteins, and the amino terminal of each subunit has a covalently linked linear tetrapyrrole (billin) chromophore [21]. PHYs could regulate plant perception of environmental light conditions such as quantity, quality and duration of light [20]. The PHY family consists of five members: PhyA, PhyB, PhyC, PhyD and PhyE. PhyA mediates irreversible photoresponses in very low and high fluence ranges (VLFR and HIR) primarily in the FR spectral region, whereas PhyB mediates the “classical” R/FR reversible responses in the low fluence range (LFR) [22]. In *A. thaliana*, the nuclear localization of PhyA and PhyB triggers a photomorphogenesis signaling pathway [23]. PhyA contributed to root elongation in F, R and blue (B) light [24]. In addition, root hair development is also mediated by PHYs, including PhyA and PhyB. Under FR exposure, PhyA could regulate root hair formation, while both PhyA and PhyB could regulate root hair formation under R illumination [24]. The *phyA-211* and *phyB-9* mutants had slow root elongation and irregular root hair formation (Table 1) [20]. Moreover, different types of PHYs showed contrary effects on lateral root formation. In *A. thaliana*, PhyA, PhyB, and PhyE induced lateral rooting, while PhyD suppressed lateral root formation [25]. PHYs also coordinate shoot/root development [26]. In *Nicotiana attenuata*, the gene expression of *NaPhyA*, *NaPhyB1*, and *NaPhyB2* was higher in the root than that in shoots (Table 1) [26]. When roots were exposed to FR irradiation, *NaPhyB1* and *NaPhyB2* mutants showed slow stalk elongation, which suggested the coordination of roots and shoots under FR exposure. PhyB also regulates root nodule formation. Moreover, plant hormones including auxin, JA and ABA could be involved in PhyA and PhyB signaling during root development [25,27,28]. The shoot-to-root transport of auxin affects lateral root formation through PhyA and PhyB signaling [25]. *PhyB* mutants showed reduced root nodule formation via jasmonic acid (JA) signaling in *Lotus japonicus* [27]. PhyB mediated ABA biosynthesis in the shoots, then the ABA signal that derived from the shoots regulated reactive oxygen species (ROS) detoxification in the roots, which could reduce the adverse effect of light on root growth [28]. PhyB could inhibit darkness-induced hypocotyl adventitious root formation by stabilizing IAA14 and suppressing ARF7 and ARF19 (Table 1) [29]. Hence, PHYs have a widespread impact in rooting.

CRYs including CRY1, CRY2 and CRY3 are blue-light (BL) receptors, which regulate different phases during plant growth and development. They are flavoproteins, which have significant homology to photolyases. CRYs functioned by transducing BL energy into a signal that can be recognized by the cellular signaling machinery [30]. In *A. thaliana*, hypocotyl phototropism is able to be regulated by CRYs. The *PhyA cry1 cry2* mutant had a defect in phototropism (Table 1) [31]. CRY1 inhibited the hypocotyl elongation [32]. Moreover, CRYs contribute to root growth [30,33]. The *PhyA,B/cry1,2* mutant had shorter roots both in light and darkness compared with WT (Table 1) [33]. The root of *cry1* mutant seedlings decreased under BL, contrary to *cry2*, which suggested that *cry1* and *cry2* antagonistically regulate primary root elongation in *A. thaliana* (Table 1) [30]. Under white light illumination, the *cry1/2* double mutant had retarded root growth in tomato and *A. thaliana* (Table 1) [34,35]. Additionally, the role of CRYs in lateral root development has been reported [36]. Under BL for 12 d, the *cry1* mutant had a significantly enhanced lateral root number, while the lateral root number in CRY1ox was significantly decreased compared with the WT [36], indicating the negative role of CRY1 in lateral rooting. CRY1 is also involved in the nodulation of roots [37]. In soybean (*Glycine max* L. Merr.), GmCRY1 was necessary for nodulation in a dark environment and inhibited the effect of BL on nodulation (Table 1) [38].

PHOT receptor kinases, which consist of a flavin mononucleotide (FMN), play important roles in promoting plant growth by controlling light-capturing processes such as phototropism [38]. PHOT1 is the primary phototropic receptor and functions over a wide range of fluence rates, whereas PHOT2 controls phototropism and the chloroplast avoidance response at higher fluence rates [39]. In *Arabidopsis*, root phototropism was regulated by PHOT1, which was active at a wide range of BL intensities (Table 1) [40]. Red light (RL) receptor photosensitivity is involved in the regulation of phototropism. Plant primary roots not only displayed blue (and white) light-induced negative phototropism, but also showed a weak red-light-mediated positive phototropic bending response in the primary root [23]. PHOT1 induced phototropism in low-flux BL. However, in the dark, it had no effect on geotropism [41]. Hypocotyl phototropism is redundantly mediated by PHOT1 and PHOT2. As light intensity increases, PHOT1 prevents the PHOT2-mediated response. In WT, and *phot1*, *phot2* and *phot1 phot2* mutants, irradiation with BL for 3 h induced *root phototropism 2* (*RPT2*) transcript levels [4]. The hypocotyl of *phot1* mutants could bend under intermediate-intensity blue light, while this effect was inhibited in the *phot1 chloroplast accumulation response 1* (*jac1*) mutant [4]. Compared with the *phot1* mutant, the *phot1 rpt2* double mutant also showed enhanced phototaxis (Table 1) [4]. Hence, PHOT2-induced hypocotyl phototropism was negatively regulated by JAC1 and RPT2.

UVR8, as a UV-B photoreceptor, is involved in UV-B photomorphogenesis and acclimation [42]. Compared with other photoreceptors, UVR8 does not combine with additional chromophores to absorb light. In fact, UVR8 directly absorbs UV-B light via its inherent tryptophan residues Trp285 and Trp337 [43]. Under low-fluence UV-B irradiation, over-expressed UVR8 plants were more dwarfed and showed a shorter hypocotyl length than the WT in *Arabidopsis* [44]. The primary root length and lateral root density in *UVR8* overexpression plants decreased compared with the WT, indicating the negative role of UVR8 in root development [44]. Moreover, UVR8 could interact with other transcription factors (TFs), such as MYB73/MYB77, in rooting. Yang et al. [12] found that UVR8 inhibited lateral root development and partly controlled hypocotyl elongation by regulating MYB73/MYB77 in a UV-B-dependent manner (Table 1). However, whether UVR8 regulates the development of root hair or adventitious roots remains unknown.

### 2.3. Key Components in Response to Light in Root Development

In addition to photoreceptors, many other components, including phytochrome-interacting factors (PIFs), constitutive photo-morphorgenic 1 (COP1), HY5, cryptochrome-interacting basic helix–loop-helixes (CIBs), and MYB73/MYB77, also play key roles in light responses. PIFs, as basic helix–loop-helix (bHLH) TFs subfamily 15, amass in the dark and boost skotomorphogenesis [45]. The PIFs family comprises PIF1, PIF2, PIF3, PIF4, PIF5 and PIF6. Among them, PIF3 could interact with PhyA and PhyB [46]. Bai et al. (2014) [46] demonstrated that PIF3 participated in the repression of nitric oxide (NO) in rooting under continuous white light (approximately 100 μmol m^−2^ s^−1^) in *A. thaliana* (Table 1). The root length of *A. thaliana* was obviously inhibited with the increase in NO donor sodium nitroprusside (SNP) concentration. However, the over-expression of *PIF3* was able to partially alleviate the inhibition of NO on root growth. In the phytochrome signal pathway, NO enhanced the accumulation of PhyB upstream of PIF3. In darkness, the inhibitory effect of NO on root growth was less influenced by PIF3 (Table 1) [47]. Under aluminium (Al) stress, PIF4 promoted primary root growth via the mediating auxin signal in the root apex transition zone in *Arabidopsis* [47]. PIFs were also reported to regulate hypocotyl elongation and adventitious rooting. Under short-day conditions, a deficiency of *PIF1, PIF3, PIF4 and PIF5* resulted in a shorter hypocotyl in *Arabidopsis* seedlings, suggesting the positive role of PIF1, PIF3, PIF4 and PIF5 in inducing hypocotyl elongation (Table 1) [48]. Both PIF4 and PIF5 promoted hypocotyl elongation by controlling BL [49]. Furthermore, PIF1 acted as the upstream of HomeoBox 1 (AtHB1) to promote hypocotyl elongation in *A. thaliana* (Table 1) [50]. In darkness, the adventitious root number in plants with the *PIF1*, *PIF2*, *PIF3*, *PIF4* and *PIF5* genes knocked out was significantly decreased, which suggested that PIFs were positive regulators in adventitious root formation (Table 1) [51]. In addition to the auxin signal, other phytohormone signal transductions could also regulate PIFs-induced hypocotyl elongation. Gibberellins (GAs) regulated sucrose-induced hypocotyl elongation in the dark [52]. In darkness, PIF1, PIF3, PIF4 and PIF5 had a positive role in sucrose-induced hypocotyl elongation [52]. Sucrose could increase the transcript level of *PIF1*, *PIF3*, *PIF4*, and *PIF5* in a gibberellin (GA)-dependent manner in the dark, indicating that PIFs could act as the downstream in sucrose and GAs-induced hypocotyl elongation [53]. PIF4 and PIF5 acted as the downstream of auxin and the ETH signaling pathway to mediate hypocotyl length under antiphase light–temperature cycles [54].

The E3 ubiquitin ligase COP1 consists of an N-terminal Really Interesting New Gene (RING) domain, a central coil–coil (CC) domain and a C-terminal WD40 repeat domain [16]. COP1 could interact with the inhibitor of PhyAs SPA and other components of the E3 ubiquitin ligase complex, and finally spurred to be degraded by the 26S proteasome [54]. COP1 plays a key role in root growth and development. The root length of *cop1-4* and *cop1-6* mutants was decreased in dark condition whereas the root length of these mutants was enhanced in light condition compared with the WT [55]. Hence, COP1 plays different roles under different conditions.

HY5, as a bZIP TF, is a positive regulator of photomorphogenesis [56]. Chen et al. (2016) [57] demonstrated that HY5 could regulate light-promoted root growth and NO_3_^−^ uptake. HY5 mediated the absorption of NO_3_^−^ by the roots in two ways. In one, the light irradiation of the shoot induced the shoot-to-root translocation of HY5, and activated the transcript level of *HY5* in *A. thaliana* [57]. Then, HY5 in the roots promoted the transcript level of the NO_3_^−^ transporter *NRT2.1* to regulate NO_3_^−^ absorption and root growth (Table 1) [57]. In the other, HY5 was helpful in maintaining the steady balance of carbon (C) and nitrogen (N) metabolism in response to ambient light conditions [57]. The transcript level of *HY5* in roots was induced by continuous white light (120 µmol m^−2^ s^−1^) [56]. However, the lack of *HY5* partially eliminated the inhibition of root growth under direct light (Table 1) [56], which suggested that *HY5* could regulate root development in response to light.

Additionally, auxin-responsive TFs MYB73/MYB77 could mediate lateral root development [12]. MYB73/MYB77 was involved in the growth of lateral roots under UV-B, and was located downstream of UVR8. The interaction between UVR8 and the DNA-binding domain of MYB73/MYB77 inhibited the DNA binding ability of MYB73/MYB77, and then inhibited its activation of auxin-related genes, thus regulating lateral rooting under UV-B in *Arabidopsis* (Table 1) [12].

**Table 1 ijms-24-05253-t001:** The photoreceptors and related genes involved in root development.

Photoreceptors	Genes	Response to Light	Species	Function	References
phyA	-	far red (FR)	*Arabidopsis thaliana*	root hair formation	[24]
phyA and phyB	-	red light (RL)	*A. thaliana*	root hair formation	[24]
phyA and phyB	*-*	-	*A. thaliana*	root elongation and irregular root hair formation	[20]
phyA, phyB, and phyE	-	-	*A. thaliana*	lateral root formation	[28]
phyD	-	-	*A. thaliana*	lateral root formation	[25]
PhyA, PhyB1, and PhyB2	*NaPhyA*, *NaPhyB1*, and *NaPhyB2*	-	*Nicotiana attenuata*	shoot-root development	[26]
phyA and phyB	-	-	*A. thaliana*	lateral root formation	[25]
*PhyB*	*-*	-	*Lotus japonicus*	root nodule formation	[27]
PhyB	*-*	-	*A. thaliana*	root growth	[28]
PhyB	*IAA14*, *ARF7* and *ARF19*		*A. thaliana*	adventitious root formation	[29]
CRY1 and CRY2	-	blue light (BL)	*A. thaliana*	primary root elongation	[30]
CRY1 and CRY2	-	white light	*A. thaliana*	primary root elongation	[34]
CRY1 and CRY2	-	white light	*Solanum lycopersicum* L.	primary root elongation	[35]
CRY1	-	BL	*A. thaliana*	lateral root formation	[36]
CRY1	-	BL	*Glycine max* L. Merr.	root nodulation	[37]
PHOT1	-	BL	*A. thaliana*	root phototropism	[40]
PHOT1 and PHOT2	*RPT2* and *JAC1*	BL	*A. thaliana*	hypocotyl phototropism	[4]
*UVR8*	-	low-fluence UV-B	*A. thaliana*	hypocotyl development	[44]
UVR8	MYB73/MYB77	UV-B-dependent manner	*A. thaliana*	lateral root development and hypocotyl elongation	[44]
PIF3	-	white light	*A. thaliana*	primary root development	[46]
PIF4	-	-	*A. thaliana*	primary root growth	[47]
PIF5	-	BL	*A. thaliana*	hypocotyl elongation	[48]
PIF1	*HB1*	-	*A. thaliana*	hypocotyl elongation	[50]
PIF1, PIF2, PIF3, PIF4 and PIF5	-	darkness	*A. thaliana*	adventitious root formation	[56]
COP1	-	darkness	*A. thaliana*	primary root length	[55]
HY5	*HY5* and *NRT2.1*	-	*A. thaliana*	root growth	[57]
HY5	*HY5*	white light	*A. thaliana*	root growth	[56]

## 3. Light Regulates Root Growth and Development via the Auxin-Signaling Transduction Pathway

### 3.1. Primary Root, Root Hair and Growth and Development

Primary roots can respond to various environmental stimuli by regulating the auxin-signaling pathway under light illumination in early seedlings. The over-expression of miR775 (miR775OX) in *A. thaliana* significantly enhanced primary root growth, with an increased transcript level of auxin biosynthesis- and transport-related genes *PIN- formed* (*PIN)1*, *PIN2*, *AUXR1*, *YUC1* and *YUC4* (Table 2, Figure 1) [58]. When plants were grown in total darkness for 5 d and then transferred to light conditions, the expression of the light response gene *HY5* was increased with the passage of time, indicating the positive role of miR775 in light response (Table 2, Figure 1) [58]. Over-expressing miR775 increased the chlorophyll accumulation (Table 2, Figure 1) [58]. Therefore, auxin could modulate primary root growth by regulating miR775, and miR775 might also regulate auxin synthesis and transport in *A. thaliana*. Furthermore, it was positively regulated by light-mediated TF HY5. Additionally, the deficiency of MEDIATOR18 (MED18) resulted in shorter primary roots under light and larger primary root elongation in darkness, which suggested that the phenotype of the *med18* root phenotype could be induced by light (Table 2, Figure 1) [59]. The expression of auxin responsive *DR5: GFP* of primary root in deficiency of MED18 seeding was increased (Figure 1) [59]. Hence, the loss of MED18 was an important component for auxin-regulated root development.

Root hairs maximize nutrient uptake and plant productivity, and rely on the environmental signals transmitted through the plant to the root tip. With direct root illumination, PIN2 expressing *eir1-4 PIN2: PIN2: VEN* mutants enhanced the number of root hairs in light-grown roots (LGRs) compared with dark-grown roots (DGRs), suggesting that root hair emergence required auxin with direct root light [60]. Furthermore, with the treatment of 1 μm auxin indole-3-acetic acid (IAA), the levels of GUS (*PromiR775: GUS*) and GFP (*PromiR775: GFP*) in transgenic lines were increased [58]. GUS expression was consistent with GUS activity, indicating crosstalk between auxin and miR775. The miR775OX lines increased the relative transcript levels of root hair developmental genes, including ROOT HAIR-DEFECTIVE LIKE 2 (RSL2), ROOT HAIR DEFECTIVE LIKE 4 (RSL4) and protein phosphatase 2A (PP2A), as well as auxin biosynthesis- and signaling-related genes such as *PIN1*, *PIN2*, *AUXR1*, *YUC1* and *YUC4*, and the light-signaling-associated gene *HY5* (Table 2, Figure 1) [58].

### 3.2. Lateral Root and Adventitious Root Growth and Development

The lateral root is an important organ for enhancing the absorption area of root systems and foraging for nutrients. Under light conditions, IAA at 10–270 nM removed a lateral root formation in the primordial formation region, which was related to the severe inhibition of root elongation [61]. In the absence of light, seedlings were induced to exhibit periodic lateral rooting by NAA, but not by IAA. A precursor of IAA biosynthesis, tryptophan, could not induce lateral root formation in dark-grown seedlings [61], indicating that the conversion of tryptophan into IAA requires light. In tobacco (*Nicotiana tabacum* L.) treated with RL, the IAA concentration in the lateral roots was higher than that in the leaves, indicating that RL could accelerate the transportation of auxin from the leaves to the roots (Table 2) [62]. RL promoted the transcript level of *PIN3* at the junction of root/shoot, while BL decreased the expression level of *PIN1*, *PIN3* and *PIN4*, indicating that the PIN1, PIN3 and PIN4 genes might play a key role in the regulation of auxin transport by RL and BL (Table 2, Figure 1) [62]. Supplementing white light with FR light decreased lateral root density in *A. thaliana* [10]. Under FR light illumination, HY5 was accumulated in lateral root primordia (LRPs) and the lateral root primordium epithelium [10]. However, the protein expression of auxin transporters, including PIN3 and LIKE AUX3 (LAX3), was declined in the cortex cells overlaying the LRP, and HY5 repressed the transcript level of auxin signaling-related genes such as *auxin response factor 19* (*ARF19*), *PIN3* and *LAX3*, indicating that HY5 could mediate lateral root germination through the auxin-signaling pathway (Table 2, Figure 1) [10]. Under UV-B irradiation, UVR8 decreased the numbers of emerged primordia at stage 7 in *A. thaliana* [12]. However, UVR8 did not show a negative effect on the numbers of emerged primordia at other stages, suggesting that UV-B could inhibit the lateral root growth in a UVR8-dependent manner. The transcript levels of auxin-responsive genes, including *HAT2*, *SUAR23*, *IAA29*, *SAUR28*, *SAUR68*, *SAUR-like* and *SAUR-like-3*, could be antagonistically regulated by auxin and UV-B irradiation (Table 2) [12]. Light responsive TF MYB73/77 could regulate lateral rooting via acting as the downstream of UVR8 (Table 2, Figure 1). UVR8 activated by UV-B inhibited lateral root growth and the transcription of the MYB73/MYB77 target by inhibiting the DNA-binding activity of MYB73/MYB77 [12].

Adventitious rooting is an important asexual propagation, which could contribute to expanding the absorption area of plant roots [63]. Constant light and a 10 μM IBA treatment for 4 d increased the adventitious root length in *Eucalyptus saligna* and *E*. *globulus* [64]. BL positively induced adventitious root formation [63]. Under BL illumination, PHOT1 and PHOT2 enhanced the number of adventitious roots and the density of adventitious root primordia (Table 2, Figure 1) [63], suggesting that the BL receptor PHOT1/PHOT2 might be involved in the induction of adventitious rooting under BL conditions. The IAA content in BL illumination treatment was higher than that in darkness [63]. PIN3 increased the number and density of adventitious root primordia, and the total number and density of adventitious roots. In addition, dark–light conversion induced adventitious rooting in *A. thaliana* (Table 2, Figure 1) [63], indicating that PIN3 could be involved in BL-induced adventitious root formation. Therefore, BL could stimulate adventitious rooting by enhancing the gene expression of *PHOT1/2*, thus increasing auxin transport and homeostasis mediated by PIN3.

### 3.3. Rhizoid, Seminal and Crown Root Development

Rhizoids are multicellular filamentous cells. In *Physcomitrella patens*, rhizoid development was positively triggered via regulating the transcript levels of the *P. patens ROOTHAIR DEFECTIVE SIX-LIKE 1* (*PpRSL1*) and *PpRSL2* genes. RSL as a TF of bHLH was associated with auxin signaling during rhizoid development [65]. CRYs negatively regulated the expression of the auxin-triggered genes *PpIAA1* and *PpGH3L1* (Table 2, Figure 1) [66]. Auxin treatment promoted rhizome development in moss, while CRY negatively regulated auxin-induced genes, indicating that CRY could regulate rhizome development by mediating the auxin signaling pathway [67].

Durative white light repressed the growth of the seminal roots needed for the healthy survival of seedlings, but it induced the emergence of crown roots in the adult stage in rice (*Oryza sativa* L.). During the development of seminal roots, the very low-fluence responses (VLFR) and low-fluence responses (LFR) were controlled by PhyA and PhyB (Table 2, Figure 1) [68]. Moreover, the light-induced root morphology was different among diverse rice varieties, with the indica and japonica varieties showing different responses to auxin during root development [69]. The transcript level of the *Oryza sativa root architecture associated 1* (*OsRAA1*) gene in the Taichung Native 1 (TCN1) seed root was induced by light in rice. The constant white light induced the seminal roots of TCN1 to become shorter and wavier [68], which suggested that seminal root development was dependent on the increase in auxin concentration and its polar transport mediated by light.

**Table 2 ijms-24-05253-t002:** Crosstalk between light and auxin signaling in root development.

Light Treatment	Genes/Proteins	Species	Function	References
darkness to light	miR775, *PIN1*, *PIN2*, *AUXR1*, *YUC1* and *YUC4*, *HY5*	*Arabidopsis thaliana*, (*A. thaliana*)	primary root growth	[58]
light	MED18	*A. thaliana*	primary root elongation	[59]
direct light	PIN2	*A. thaliana*	root hair formation	[60]
light	miR775, RSL2, RSL4, PP2A, HY5, *PIN1*, *PIN2*, *AUXR1*, *YUC1* and *YUC4*	*A. thaliana*	root hair formation	[58]
red light (RL)	*PIN3*	*A. thaliana*	lateral root development	[62]
blue light (BL)	PIN1, PIN3 and PIN4	*A. thaliana*	lateral root development	[62]
white light with FR light	-	*A. thaliana*	decreased lateral root density	[10]
far red (FR) light	HY5, *ARF19*, *PIN3* and *LAX3*	*A. thaliana*	lateral root development	[10]
UV-B	*HAT2*, *SUAR23*, MYB73/MYB77, UVR8, *IAA29*, *SAUR28*, *SAUR68*, *SAUR-like* and *SAUR-like-3*	*A. thaliana*	lateral root growth	[16]
BL	PIN3, PHOT1 and PHOT2	*A. thaliana*	adventitious root formation	[63]
-	CRYs	*Physcomitrella patens*	rhizoid development	[67]
white light	PHYA and PHYB	*Oryza sativa* L.	seminal root development	[68]
BL	*CRY1*, *PHOT2, PIN3*	*A. thaliana*	root negative phototropism	[45]
BL	PHOT1, PIN1 and PIN2	*A. thaliana*	root negative phototropism	[46]
BL	PIN2	*A. thaliana*	root negative phototropism	[70]
white light	*ZM2G141383*	*Zea mays*	root gravitropism	[71]
-	HY5, GLK2, *IAA14*, *ARF7* and *ARF19*	*A. thaliana*	root greening	[72]
-	HY1, *HY5*, *HYH*, *AUX1*, *PIN1*, *PIN2*, *PIN3* and *PIN7*,	*A. thaliana*	lateral root branching	[73]
-	HY5, *AXR2/IAA7* and *SLR/IAA14*	*A. thaliana*	lateral root branching	[74]

## 4. Light-Regulated Tropic Movement, Root Greening and Root Branching through Auxin Signaling

### 4.1. Root Negative Phototropism

Root negative phototropism is one of the key responses of plants in adapting to environmental changes. Zhang et al. [75] reported that PIN 3, as an auxin efflux carrier, could be involved in asymmetric auxin distribution, which ultimately caused root negative phototropism. Under low fluence rates (10 mol m^−2^ s^−1^) of BL illumination, the absence of BL receptor *CRY1* and *PHOT2* genes showed normal root negative phototropism in *A. thaliana* (Table 2, Figure 1) [75]. However, knocking down the *PHOT1* gene weakened the phototropic response of roots (Table 2, Figure 1) [64], which indicated that BL-promoted root negative phototropism could depend on PHOT1. The asymmetric redistribution of auxin regulated by PHOT1 participated in root negative phototropism (Figure 1). Some proteins including PHYTOCHROME KINASE SUBSTRATE 1 (PKS1), ROOT PHOTOTROPISM 2 (RPT2) and NON-PHOTOTROPIC HYPOCOTYL 3 (NPH3) were also able to interact with PHOT1 in the negative phototropic response (Table 2, Figure 1) [70,76,77,78]. PIN3 increased the root negative phototropic response under BL [75], suggesting that PIN3 polarization was involved in the generation of asymmetric auxin distribution. When the roots were irradiated by unilateral blue light, PIN3 aggregated to the outer membrane of the columella cells, leading to the accumulation of auxin in the roots on the irradiated side. Enhancing the concentration of auxin-induced root growth on the light side, and caused the root to bend away from the light [75]. The polar distribution of root-oriented PIN3 was modulated by the brefeldin A (BFA)-sensitive transport pathway and PINOID (PID)/PROTEIN PHOSPHATASE 2A (PP2A) activity (Figure 1). Moreover, PIN1 and PIN2 could also affect BL-mediated root phototropism in *A. thaliana*. PIN1 was necessary for hypocotyl phototropic responses [79]. In this process, unilateral BL could lead to the relocation of PIN1 in hypocotyl cells [79]. From darkness to BL irradiation, the localization of PIN1 shifted from the intracellular compartment of the root to the stele cells in the basal plasma membranes [80]. The BL-regulated distribution of PIN1 induced the asymmetric distribution of auxin and root negative phototropism [80]. PHOT1, as the main blue light receptor, mediated the redistribution of PIN1 in roots [80]. BL-induced PIN1 redistribution was regulated by BFA-sensitive and GNOM-dependent transport pathways and by PID/PP2A activity (Figure 1) [80]. Hence, a BL-regulated root negative phototropic response occurred by locating PIN1 through PHOT1. PIN2 was the other important component for BL-mediated root phototropism. Under unilateral BL irradiation, the location shift of PIN2, which was controlled by BL and the BFA-sensitive pathway, changed the distribution of auxin in roots, leading to root negative phototropism (Table 2, Figure 1) [70].

### 4.2. Gravitropism/U-Turn Formation at Root Apex

Gravitropism is the movement or growth of plants affected by gravity [71]. The root shows positive geotropism. Generally speaking, the movement of gravitropism response is often from root cap to root apex [71]. In some monocot plants such as maize (*Zea mays*), the root cap needs light to generate a positive geotropic response. Under continuous white light (~36 µmol m^−2^ s^−1^) irradiation for ≥1–2 h, the content of IAA was increased in maize root tips, especially in the transition zone (Table 2, Figure 1) [71]. With IAA biosynthesis inhibitors yucasin and l-kynurenine under white light, the IAA level and root curvature in the root apex region was significantly decreased in maize root tips [72], indicating that white light could induce root apex gravitropism by increasing the IAA content. Using the isotope labeling of IAA precursor tryptophan, it was found that IAA was biosynthesized in the root apex [71]. The transcript level of IAA the biosynthesis-related *Zmyuc* gene *ZM2G141383* was up-regulated in the 0–1 mm tip region with the exposure to white light (Table 2, Figure 1) [71], implying that the accumulation of IAA in the transition zone was due to the white-light-triggered activation of the *Zmyuc* transcript level in the 0–1 mm root apex region. Therefore, the formation of the root apex U-turn might play a positive role via the increase in IAA accumulation and IAA distribution in maize roots under consistent white light.

### 4.3. Root Greening and Root Branching

Root greening is an important event during plant growth and development. Auxin prevented root greening by down-regulating the expression of *IAA14*, *auxin response factor 7* (*ARF7*) and *ARF19* (Figure 1) [72]. Moreover, this regulation by the auxin-signaling pathway was associated with light-related TF HY5 [72]. Root greening TF GOLDEN2-LIKE2 (GLK2), as the downstream of the hormone and light signal pathway, could interact with HY5 to synergistically induce root greening in *A. thaliana* (Table 2, Figure 1) [72]. Therefore, the phenomenon of plant root greening may be a result of the interaction between light and auxin. Root branching is a key factor that affects lateral root growth and development. Under continuous light (100 µmol m^−2^ s^−1^), HY1 which was produced in the pericycle cells of branches or local xylem could mediate lateral root branching (Table 2, Figure 1) [73].

HY1 induced the transcript level of *HY5* and its homolog *HYH*, and up-regulated the expression of auxin transporters such as *AUX1*, *PIN1*, *PIN2*, *PIN3* and *PIN7*, thus promoting the accumulation of auxin in the oscillation region and triggering lateral root branching (Table 2, Figure 1) [73]. Furthermore, HY1 and HY5 also regulated lateral root branching under white light. When exposed to white light, HY5 induced lateral root branching in *Arabidopsis* through enhancing the transcript level of two negative regulators of auxin signaling *AUXIN RESISTANT 2* (*AXR2*)/*INDOLE ACETIC ACID 7* (*IAA7*) and *SOLITARY ROOT* (SLR)/*IAA14* (Table 2, Figure 1) [74].

## 5. Conclusions

Light, as one of the most important environmental factors, can affect root growth and development via the auxin-signaling pathway. Over the past several decades, there has been an increasing number of studies conducted on the light-mediated root development via auxin signal. In this review, we summarized the role of light-regulated auxin signaling in root growth and development. First, some photoreceptors, such as PHYs, PHOTs and CRYs are involved in root development, for example, the primary root, root hair, adventitious root and lateral root development. Additionally, other TFs including PIFs, HY5, COP1and MYB73/77 are associated with light response during root development. Auxin, as an essential plant hormone, can interact with these photoreceptors to regulate root growth and development. For example, the BL receptor PHOT1/PHOT2 might be involved in the induction of adventitious rooting under BL conditions by increasing the transcript level of the auxin-signaling gene *PIN3*. Some auxin-signaling-related target genes, such as *PIN1*, *PIN2*, *PIN3*, *LAX3*, *ARF19*, *HAT2*, *SUAR23*, *SAUR28* and *IAA29*, have also been demonstrated in root development. Light not only affects the growth and development of primary roots, root hair, lateral roots, and adventitious roots through auxin signals, but also affects phototropism, gravitropism, root greening, and root branching. Despite continuous advancements, the molecular mechanism underlying the crosstalk between light and auxin signaling during rooting still needs to be illustrated and completed. Therefore, further research is required to investigate these mechanisms. Finally, the question of whether there any other photoreceptors that can be used to regulate root development via the auxin-signaling pathway should be considered. In future, related work should be carried out to improve our knowledge regarding the crosstalk between light and auxin signaling during rooting in plants.

## Figures and Tables

**Figure 1 ijms-24-05253-f001:**
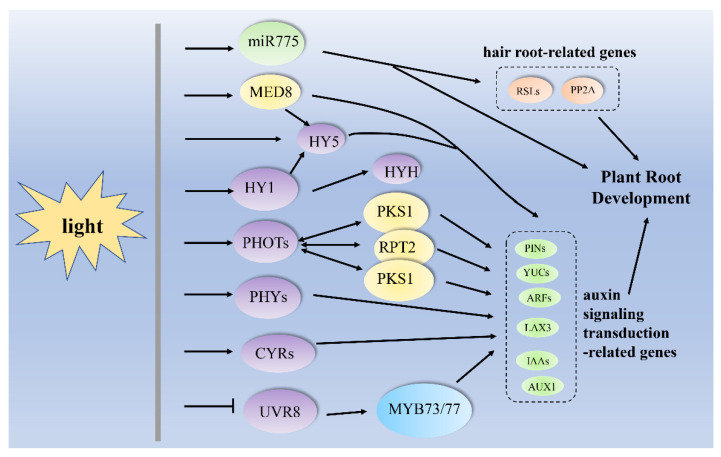
(HY5), UV resistance locus 8 (UVR8), phototropins (PHOTs), phytochromes (PHYs) and cryptochromes (CRYs) are regarded as photoreceptors with a response to light, and they act as the upstream of auxin-signaling-related genes or proteins. PHOTs could interact with PHYTOCHROME KINASE SUBSTRATE 1 (PKS1), ROOT PHOTOTROPISM 2 (RPT2) and NON-PHOTOTROPIC HYPOCOTYL 3 (NPH3), and act as the upstream of the auxin-signaling-related genes PINs. The MYB domain protein 73/77 (MYB73/MYB77) acts as the downstream of UVR8 to regulate auxin responses and lateral root growth. In the auxin-signaling transduction pathway, PIN-formed (*PINs*), *YUCs*, *auxin response factors* (*ARFs*) and LIKE AUX3 (*LAX3*) are key genes, and act as the downstream of light-response-related genes or proteins. When a plant is exposed to light, miR775 could induce the root growth and development by regulating root-hair-related genes such as ROOTHAIR DEFECTIVE SIX-LIKE (*RSLs*) and *PROTEIN PHOSPHATASE 2A* (*PP2A*). MEDIATOR18 (MED18) could promote root growth with continuous light via directly regulating auxin-signaling transduction-related genes. Moreover, MED18 triggered the transcript level of *HY5*, which led to chlorophyll accumulation. When a plant is exposed to light, HY1 induces the up-regulation of HY5 and HYH, then HY5 triggers the expression of auxin-signaling-related genes IAAs, which finally promotes root branching. Furthermore, HY1 can directly regulate root branching via enhancing the transcript level of PINs and AUX1. The arrows mean that one protein or stimuli could directly and positively act as the upstream to regulate another protein. The double arrows represent interactions between proteins. The “T” shape indicates that one stimuli could negatively regulate the protein.

## Data Availability

Not applicable.

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
