# Peer review of "The Role of Light-Regulated Auxin Signaling in Root Development"

_ijms, 2023, doi:10.3390/ijms24065253_

Round 1

Reviewer 1 Report

Yun et al summerized recent discoveries related to cross-talk between auxin and light signaling pathways in regulating root growth. I found the generalization and conclusions are generally sound although the article is not a hypothesis-driven research article, it is hard to judge its novelty, its methodology, if controls are sufficient, whether data support their conclusions and whether it fills a gap or not.

The models (Figs. 1 and 2) are easy to understand and good summerizations. However, as the authors separated the components in the root responses into different sub-figures, it would be great that if authors could consider integrating most of the components into one  figure and within one network. 

The titles for sections 3.3, 4.2 and 4.3 are ALL the SAME, despite they are on different subjects.

Subtitles for 3.1, 3.2 and 3.3 need to be more precise, the authors may delete "Crosstalk between light and auxin signaling in".

I understand that English is not the authros' native language, however, they should better avoid using "consummated" (line 448).

Line 226, it should be "Development";

Line 375, it should read as "Figure 2. Schematic ...".

The authors have included all of the necessary references covering the topic in their references. They may need to include the following paper into discussion:
Transmembrane kinase-1-mediated auxin signal regulates membrane-associated clathrin in Arabidopsis roots; DOI: 10.1111/jipb.13366

Reviewer 2 Report

This review article is very detailed and comprehensive about the role of light-regulated auxin signaling in root development, and is highly worthy of publication in IJMS. I would like only the following corrections:

1) The authors state that “We conclude that the mechanism of light-me-diated root development via auxin signaling is complex, mainly involved in the differences of plant species (line 19)”, so please indicate the plant species name in the sentences, line 45 to 60, as well. 

2)    Line 51

I couldn’t understand “cortex above LRP”. The “above” is ambiguous, so I request another precise explanation. 

3)    Line 96

inclduing --> including

4)    Line 98

mutants had slow slow root elongation --> mutants had slow root elongation

5)    Line 164

hair root --> root hair?

6)    Line 206 “HY5 could regulate light-promoted root growth and NO3- uptake, and this process was dependent on HY5.

I think “, and this process was dependent on HY5” is unnecessary.  

7)    Line 301 “Adventitious rooting is an important link of asexual propagation, which could contribute to expanding the absorption area of plant roots

Please explain why you wrote “an important link of asexual propagation”. 

8)    About sections’ titles, 3.3, 4.2, and 4.3. 

All these titles are the same. Please check it. 

Reviewer 3 Report

Yun et al. reviewed light and auxin-dependent root development in plants. Article style is ok but some sentences are hard to understand. The figures do not help to understand what the authors explain there. Please improve the points below

1. The illustrations in the figures: Please enhance contrast of letters on the objects. Use white letters on dark objects.

2. Arrows in the illustration do not make sense. What do they mean? How do the proteins/stimuli affect? Why the arrows across the membrane? Why the transporters are inside of the intracellular membrane? Please draw understandable illustrations.

3. English must be improved. I do not see the points of some sentences. For example, L341: “negative phototropism of auxin” does not make sense. Asymmetric auxin distribution causes negative phototropism. L371: “However~”, this sentence also does not make sense.

Round 2

Reviewer 3 Report

Thank you very much for your improvement of the manuscript. Now it looks very good.